# FRACTALNET:
# ULTRA-DEEP NEURAL NETWORKS WITHOUT RESIDUALS

**Gustav Larsson**
University of Chicago
larsson@cs.uchicago.edu

**Michael Maire**
TTI Chicago
mmaire@ttic.edu

**Gregory Shakhnarovich**
TTI Chicago
greg@ttic.edu

## ABSTRACT

We introduce a design strategy for neural network macro-architecture based on self-similarity. Repeated application of a simple expansion rule generates deep networks whose structural layouts are precisely truncated fractals. These networks contain interacting subpaths of different lengths, but do not include any pass-through or residual connections; every internal signal is transformed by a filter and nonlinearity before being seen by subsequent layers. In experiments, fractal networks match the excellent performance of standard residual networks on both CIFAR and ImageNet classification tasks, thereby demonstrating that residual representations may not be fundamental to the success of extremely deep convolutional neural networks. Rather, the key may be the ability to transition, during training, from effectively shallow to deep. We note similarities with student-teacher behavior and develop drop-path, a natural extension of dropout, to regularize co-adaptation of subpaths in fractal architectures. Such regularization allows extraction of high-performance fixed-depth subnetworks. Additionally, fractal networks exhibit an anytime property: shallow subnetworks provide a quick answer, while deeper subnetworks, with higher latency, provide a more accurate answer.

## 1 INTRODUCTION

Residual networks (He et al., 2016a), or ResNets, lead a recent and dramatic increase in both depth and accuracy of convolutional neural networks, facilitated by constraining the network to learn residuals. ResNet variants (He et al., 2016a;b; Huang et al., 2016b) and related architectures (Srivastava et al., 2015) employ the common technique of initializing and anchoring, via a pass-through channel, a network to the identity function. Training now differs in two respects. First, the objective changes to learning residual outputs, rather than unreferenced absolute mappings. Second, these networks exhibit a type of deep supervision (Lee et al., 2014), as near-identity layers effectively reduce distance to the loss. He et al. (2016a) speculate that the former, the residual formulation itself, is crucial.

We show otherwise, by constructing a competitive extremely deep architecture that does not rely on residuals. Our design principle is pure enough to communicate in a single word, fractal, and a simple diagram (Figure 1). Yet, fractal networks implicitly recapitulate many properties hard-wired into previous successful architectures. Deep supervision not only arises automatically, but also drives a type of student-teacher learning (Ba & Caruana, 2014; Urban et al., 2017) internal to the network. Modular building blocks of other designs (Szegedy et al., 2015; Liao & Carneiro, 2015) resemble special cases of a fractal network's nested substructure.

For fractal networks, simplicity of training mirrors simplicity of design. A single loss, attached to the final layer, suffices to drive internal behavior mimicking deep supervision. Parameters are randomly initialized. As they contain subnetworks of many depths, fractal networks are robust to choice of overall depth; make them deep enough and training will carve out a useful assembly of subnetworks.

The entirety of emergent behavior resulting from a fractal design may erode the need for recent engineering tricks intended to achieve similar effects. These tricks include residual functional forms with identity initialization, manual deep supervision, hand-crafted architectural modules, and student-teacher training regimes. Section 2 reviews this large body of related techniques. Hybrid designs could certainly integrate any of them with a fractal architecture; we leave open the question of the degree to which such hybrids are synergistic.

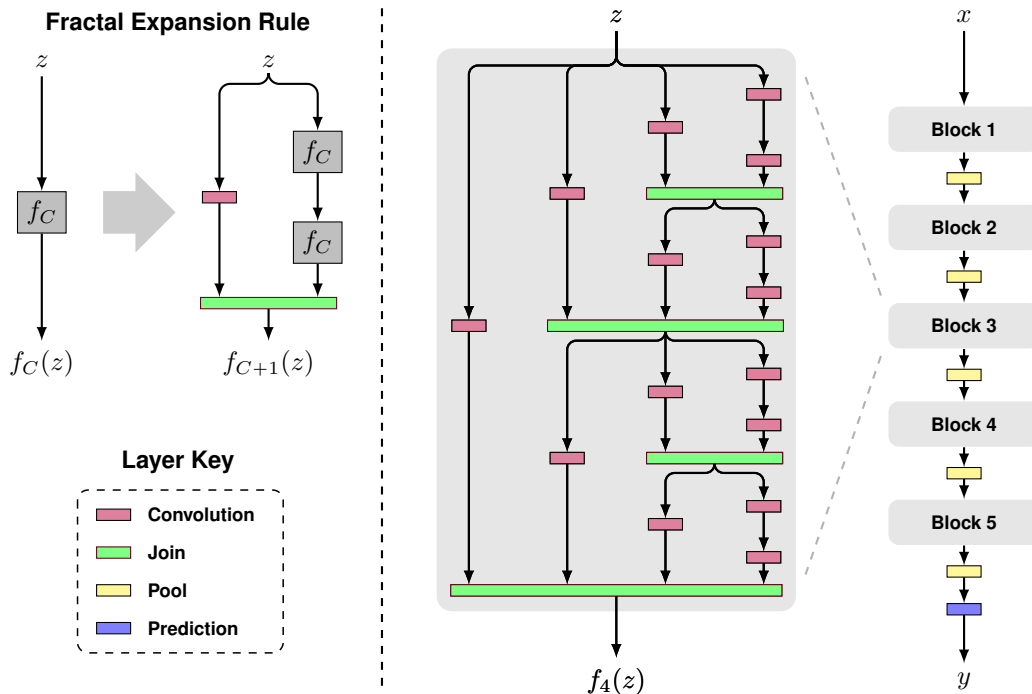

Figure 1: **Fractal architecture.** *Left:* A simple expansion rule generates a fractal architecture with $C$ intertwined columns. The base case, $f_1(z)$, has a single layer of the chosen type (*e.g.* convolutional) between input and output. Join layers compute element-wise mean. *Right:* Deep convolutional networks periodically reduce spatial resolution via pooling. A fractal version uses $f_C$ as a building block between pooling layers. Stacking $B$ such blocks yields a network whose total depth, measured in terms of convolution layers, is $B \cdot 2^{C-1}$. This example has depth 40 ($B = 5$, $C = 4$).

Our main contribution is twofold:

- We introduce FractalNet, the first simple alternative to ResNet. FractalNet shows that explicit residual learning is not a requirement for building ultra-deep neural networks.

- Through analysis and experiments, we elucidate connections between FractalNet and an array of phenomena engineered into previous deep network designs.

As an additional contribution, we develop drop-path, a novel regularization protocol for ultra-deep fractal networks. Without data augmentation, fractal networks, trained with drop-path and dropout (Hinton et al., 2012), exceed the performance of residual networks regularized via stochastic depth (Huang et al., 2016b). Though, like stochastic depth, it randomly removes macro-scale components, drop-path further exploits our fractal structure in choosing which components to disable.

Drop-path constitutes not only a regularization strategy, but also provides means of optionally imparting fractal networks with anytime behavior. A particular schedule of dropped paths during learning prevents subnetworks of different depths from co-adapting. As a consequence, both shallow and deep subnetworks must individually produce correct output. Querying a shallow subnetwork thus yields a quick and moderately accurate result in advance of completion of the full network.

Section 3 elaborates the technical details of fractal networks and drop-path. Section 4 provides experimental comparisons to residual networks across the CIFAR-10, CIFAR-100 (Krizhevsky, 2009), SVHN (Netzer et al., 2011), and ImageNet (Deng et al., 2009) datasets. We also evaluate regularization and data augmentation strategies, investigate subnetwork student-teacher behavior during training, and benchmark anytime networks obtained using drop-path. Section 5 provides synthesis. By virtue of encapsulating many known, yet seemingly distinct, design principles, self-similar structure may materialize as a fundamental component of neural architectures.

## 2 RELATED WORK

Deepening feed-forward neural networks has generally returned dividends in performance. A striking example within the computer vision community is the improvement on the ImageNet (Deng et al., 2009) classification task when transitioning from AlexNet (Krizhevsky et al., 2012) to VGG (Simonyan & Zisserman, 2015) to GoogLeNet (Szegedy et al., 2015) to ResNet (He et al., 2016a). Unfortunately, greater depth also makes training more challenging, at least when employing a first-order optimization method with randomly initialized layers. As the network grows deeper and more non-linear, the linear approximation of a gradient step becomes increasingly inappropriate. Desire to overcome these difficulties drives research on both optimization techniques and network architectures.

On the optimization side, much recent work yields improvements. To prevent vanishing gradients, ReLU activation functions now widely replace sigmoid and tanh units (Nair & Hinton, 2010). This subject remains an area of active inquiry, with various tweaks on ReLUs, *e.g.* PReLUs (He et al., 2015), and ELUs (Clevert et al., 2016). Even with ReLUs, employing batch normalization (Ioffe & Szegedy, 2015) speeds training by reducing internal covariate shift. Good initialization can also ameliorate this problem (Glorot & Bengio, 2010; Mishkin & Matas, 2016). Path-SGD (Neyshabur et al., 2015) offers an alternative normalization scheme. Progress in optimization is somewhat orthogonal to our architectural focus, with the expectation that advances in either are ripe for combination.

Notable ideas in architecture reach back to skip connections, the earliest example of a nontrivial routing pattern within a neural network. Recent work further elaborates upon them (Maire et al., 2014; Hariharan et al., 2015). Highway networks (Srivastava et al., 2015) and ResNet (He et al., 2016a;b) offer additional twists in the form of parameterized pass-through and gating. In work subsequent to our own, Huang et al. (2016a) investigate a ResNet variant with explicit skip connections. These methods share distinction as the only other designs demonstrated to scale to hundreds of layers and beyond. ResNet's building block uses the identity map as an anchor point and explicitly parameterizes an additive correction term (the residual). Identity initialization also appears in the context of recurrent networks (Le et al., 2015). A tendency of ResNet and highway networks to fall-back to the identity map may make their effective depth much smaller than their nominal depth.

Some prior results hint at what we experimentally demonstrate in Section 4. Namely, reduction of effective depth is key to training extremely deep networks; residuals are incidental. Huang et al. (2016b) provide one clue in their work on stochastic depth: randomly dropping layers from ResNet during training, thereby shrinking network depth by a constant factor, provides additional performance benefit. We build upon this intuition through drop-path, which shrinks depth much more drastically.

The success of deep supervision (Lee et al., 2014) provides another clue that effective depth is crucial. Here, an auxiliary loss, forked off mid-level layers, introduces a shorter path during backpropagation. The layer at the fork receives two gradients, originating from the main loss and the auxiliary loss, that are added together. Deep supervision is now common, being adopted, for example, by GoogLeNet (Szegedy et al., 2015). However, irrelevance of the auxiliary loss at test time introduces the drawback of having a discrepancy between the actual objective and that used for training.

Exploration of the student-teacher paradigm (Ba & Caruana, 2014) illuminates the potential for interplay between networks of different depth. In the model compression scenario, a deeper network (previously trained) guides and improves the learning of a shallower and faster student network (Ba & Caruana, 2014; Urban et al., 2017). This is accomplished by feeding unlabeled data through the teacher and having the student mimic the teacher's soft output predictions. FitNets (Romero et al., 2015) explicitly couple students and teachers, forcing mimic behavior across several intermediate points in the network. Our fractal networks capture yet another alternative, in the form of implicit coupling, with the potential for bidirectional information flow between shallow and deep subnetworks.

Widening networks, by using larger modules in place of individual layers, has also produced performance gains. For example, an Inception module (Szegedy et al., 2015) concatenates results of convolutional layers of different receptive field size. Stacking these modules forms the GoogLeNet architecture. Liao & Carneiro (2015) employ a variant with maxout in place of concatenation. Figure 1 makes apparent our connection with such work. As a fractal network deepens, it also widens. Moreover, note that stacking two 2D convolutional layers with the same spatial receptive field (*e.g.* $3 \times 3$) achieves a larger ($5 \times 5$) receptive field. A horizontal cross-section of a fractal network is reminiscent of an Inception module, except with additional joins due to recursive structure.

## 3   FRACTAL NETWORKS

We begin with a more formal presentation of the ideas sketched in Figure 1. Convolutional neural networks serve as our running example and, in the subsequent section, our experimental platform. However, it is worth emphasizing that our framework is more general. In principle, convolutional layers in Figure 1 could be replaced by a different layer type, or even a custom-designed module or subnetwork, in order to generate other fractal architectures.

Let $C$ denote the index of the truncated fractal $f_C(\cdot)$. Our network's structure, connections and layer types, is defined by $f_C(\cdot)$. A network consisting of a single convolutional layer is the base case:

$$f_1(z) = \text{conv}(z) \tag{1}$$

We define successive fractals recursively:

$$f_{C+1}(z) = [(f_C \circ f_C)(z)] \oplus [\text{conv}(z)] \tag{2}$$

where $\circ$ denotes composition and $\oplus$ a join operation. When drawn in the style of Figure 1, $C$ corresponds to the number of columns, or width, of network $f_C(\cdot)$. Depth, defined to be the number of conv layers on the longest path between input and output, scales as $2^{C-1}$. Convolutional networks for classification typically intersperse pooling layers. We achieve the same by using $f_C(\cdot)$ as a building block and stacking it with subsequent pooling layers $B$ times, yielding total depth $B \cdot 2^{C-1}$.

The join operation $\oplus$ merges two feature blobs into one. Here, a blob is the result of a conv layer: a tensor holding activations for a fixed number of channels over a spatial domain. The channel count corresponds to the size of the filter set in the preceding conv layer. As the fractal is expanded, we collapse neighboring joins into a single join layer which spans multiple columns, as shown on the right side of Figure 1. The join layer merges all of its input feature blobs into a single output blob.

Several choices seem reasonable for the action of a join layer, including concatenation and addition. We instantiate each join to compute the element-wise mean of its inputs. This is appropriate for convolutional networks in which channel count is set the same for all conv layers within a fractal block. Averaging might appear similar to ResNet's addition operation, but there are critical differences:

- ResNet makes clear distinction between pass-through and residual signals. In FractalNet, no signal is privileged. Every input to a join layer is the output of an immediately preceding conv layer. The network structure alone cannot identify any as being primary.

- Drop-path regularization, as described next in Section 3.1, forces each input to a join to be individually reliable. This reduces the reward for even implicitly learning to allocate part of one signal to act as a residual for another.

- Experiments show that we can extract high-performance subnetworks consisting of a single column (Section 4.2). Such a subnetwork is effectively devoid of joins, as only a single path is active throughout. They produce no signal to which a residual could be added.

Together, these properties ensure that join layers are not an alternative method of residual learning.

### 3.1   REGULARIZATION VIA DROP-PATH

Dropout (Hinton et al., 2012) and drop-connect (Wan et al., 2013) modify interactions between sequential network layers in order to discourage co-adaptation. Since fractal networks contain additional macro-scale structure, we propose to complement these techniques with an analogous coarse-scale regularization scheme.

Figure 2 illustrates drop-path. Just as dropout prevents co-adaptation of activations, drop-path prevents co-adaptation of parallel paths by randomly dropping operands of the join layers. This discourages the network from using one input path as an anchor and another as a corrective term (a configuration that, if not prevented, is prone to overfitting). We consider two sampling strategies:

- **Local**: a join drops each input with fixed probability, but we make sure at least one survives.

- **Global**: a single path is selected for the entire network. We restrict this path to be a single column, thereby promoting individual columns as independently strong predictors.

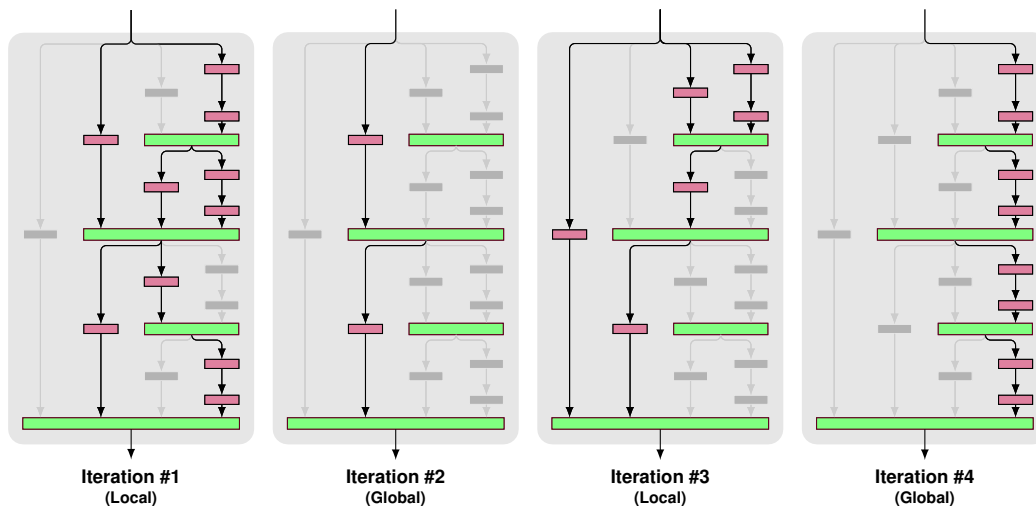

Figure 2: **Drop-path.** A fractal network block functions with some connections between layers disabled, provided some path from input to output is still available. Drop-path guarantees at least one such path, while sampling a subnetwork with many other paths disabled. During training, presenting a different active subnetwork to each mini-batch prevents co-adaptation of parallel paths. A global sampling strategy returns a single column as a subnetwork. Alternating it with local sampling encourages the development of individual columns as performant stand-alone subnetworks.

As with dropout, signals may need appropriate rescaling. With element-wise means, this is trivial; each join computes the mean of only its active inputs.

In experiments, we train with dropout and a mixture model of 50% local and 50% global sampling for drop-path. We sample a new subnetwork each mini-batch. With sufficient memory, we can simultaneously evaluate one local sample and all global samples for each mini-batch by keeping separate networks and tying them together via weight sharing.

While fractal connectivity permits the use of paths of any length, global drop-path forces the use of many paths whose lengths differ by orders of magnitude (powers of 2). The subnetworks sampled by drop-path thus exhibit large structural diversity. This property stands in contrast to stochastic depth regularization of ResNet, which, by virtue of using a fixed drop probability for each layer in a chain, samples subnetworks with a concentrated depth distribution (Huang et al., 2016b).

Global drop-path serves not only as a regularizer, but also as a diagnostic tool. Monitoring performance of individual columns provides insight into both the network and training mechanisms, as Section 4.3 discusses in more detail. Individually strong columns of various depths also give users choices in the trade-off between speed (shallow) and accuracy (deep).

## 3.2 DATA AUGMENTATION

Data augmentation can reduce the need for regularization. ResNet demonstrates this, achieving 27.22% error rate on CIFAR-100 with augmentation compared to 44.76% without (Huang et al., 2016b). While augmentation benefits fractal networks, we show that drop-path provides highly effective regularization, allowing them to achieve competitive results even without data augmentation.

## 3.3 IMPLEMENTATION DETAILS

We implement FractalNet using Caffe (Jia et al., 2014). Purely for convenience, we flip the order of pool and join layers at the end of a block in Figure 1. We pool individual columns immediately before the joins spanning all columns, rather than pooling once immediately after them.

We train fractal networks using stochastic gradient descent with momentum. As now standard, we employ batch normalization together with each conv layer (convolution, batch norm, then ReLU).

| Method | C100 | C100+ | C100++ | C10 | C10+ | C10++ | SVHN |
|---|---|---|---|---|---|---|---|
| Network in Network (Lin et al., 2013) | 35.68 | - | - | 10.41 | 8.81 | - | 2.35 |
| Generalized Pooling (Lee et al., 2016) | 32.37 | - | - | 7.62 | 6.05 | - | 1.69 |
| Recurrent CNN (Liang & Hu, 2015) | 31.75 | - | - | 8.69 | 7.09 | - | 1.77 |
| Multi-scale (Liao & Carneiro, 2015) | 27.56 | - | - | 6.87 | - | - | 1.76 |
| FitNet Romero et al. (2015) | - | 35.04 | - | - | 8.39 | - | 2.42 |
| Deeply Supervised (Lee et al., 2014) | - | 34.57 | - | 9.69 | 7.97 | - | 1.92 |
| All-CNN (Springenberg et al., 2014) | - | 33.71 | - | 9.08 | 7.25 | 4.41 | - |
| Highway Net (Srivastava et al., 2015) | - | 32.39 | - | - | 7.72 | - | - |
| ELU (Clevert et al., 2016) | - | 24.28 | - | - | 6.55 | - | - |
| Scalable BO (Snoek et al., 2015) | - | - | 27.04 | - | - | 6.37 | 1.77 |
| Fractional Max-Pool (Graham, 2014) | - | - | 26.32 | - | - | 3.47 | - |
| FitResNet (Mishkin & Matas, 2016) | - | 27.66 | - | - | 5.84 | - | - |
| ResNet (He et al., 2016a) | - | - | - | - | 6.61 | - | - |
| ResNet by (Huang et al., 2016b) | 44.76 | 27.22 | - | 13.63 | 6.41 | - | 2.01 |
| Stochastic Depth (Huang et al., 2016b) | 37.80 | 24.58 | - | 11.66 | 5.23 | - | 1.75 |
| Identity Mapping (He et al., 2016b) | - | 22.68 | - | - | 4.69 | - | - |
| ResNet in ResNet (Targ et al., 2016) | - | 22.90 | - | - | 5.01 | - | - |
| Wide (Zagoruyko & Komodakis, 2016) | - | 20.50 | - | - | 4.17 | - | - |
| DenseNet-BC (Huang et al., 2016a)[1] | 19.64 | 17.60 | - | 5.19 | 3.62 | - | 1.74 |
| FractalNet (20 layers, 38.6M params) | 35.34 | 23.30 | 22.85 | 10.18 | 5.22 | 5.11 | 2.01 |
| + drop-path + dropout | 28.20 | 23.73 | 23.36 | 7.33 | 4.60 | 4.59 | 1.87 |
| ↳ deepest column alone | 29.05 | 24.32 | 23.60 | 7.27 | 4.68 | 4.63 | 1.89 |
| FractalNet (40 layers, 22.9M params)[2] | - | 22.49 | 21.49 | - | 5.24 | 5.21 | - |

Table 1: **CIFAR-100/CIFAR-10/SVHN.** We compare test error (%) with other leading methods, trained with either no data augmentation, translation/mirroring (+), or more substantial augmentation (++). Our main point of comparison is ResNet. We closely match its benchmark results using data augmentation, and outperform it by large margins without data augmentation. Training with drop-path, we can extract from FractalNet single-column (plain) networks that are highly competitive.

## 4 EXPERIMENTS

The CIFAR, SVHN, and ImageNet datasets serve as testbeds for comparison to prior work and analysis of FractalNet's internal behavior. We evaluate performance on the standard classification task associated with each dataset. For CIFAR and SVHN, which consist of $32 \times 32$ images, we set our fractal network to have 5 blocks ($B = 5$) with $2 \times 2$ non-overlapping max-pooling and subsampling applied after each. This reduces the input $32 \times 32$ spatial resolution to $1 \times 1$ over the course of the entire network. A softmax prediction layer attaches at the end of the network. Unless otherwise noted, we set the number of filter channels within blocks 1 through 5 as $(64, 128, 256, 512, 512)$, mostly matching the convention of doubling the number of channels after halving spatial resolution.

For ImageNet, we choose a fractal architecture to facilitate direct comparison with the 34-layer ResNet of He et al. (2016a). We use the same first and last layer as ResNet-34, but change the middle of the network to consist of 4 blocks ($B = 4$), each of 8 layers ($C = 4$ columns). We use a filter channel progression of $(128, 256, 512, 1024)$ in blocks 1 through 4.

### 4.1 TRAINING

For experiments using dropout, we fix drop rate per block at $(0\%, 10\%, 20\%, 30\%, 40\%)$, similar to Clevert et al. (2016). Local drop-path uses $15\%$ drop rate across the entire network.

---

[1] Densely connected networks (DenseNets) are concurrent work, appearing subsequent to our original arXiv paper on FractalNet. A variant of residual networks, they swap addition for concatenation in the residual functional form. We report performance of their 250-layer DenseNet-BC network with growth rate $k = 24$.

[2] This deeper (4 column) FractalNet has fewer parameters. We vary column width: $(128, 64, 32, 16)$ channels across columns initially, doubling each block except the last. A linear projection temporarily widens thinner columns before joins. As in Iandola et al. (2016), we switch to a mix of $1 \times 1$ and $3 \times 3$ convolutional filters.

| Method | Top-1 (%) | Top-5 (%) |
|---|---|---|
| VGG-16 | 28.07 | 9.33 |
| ResNet-34 C | 24.19 | 7.40 |
| FractalNet-34 | 24.12 | 7.39 |

Table 2: **ImageNet** (validation set, 10-crop).

| Cols. | Depth | Params. | Error (%) |
|---|---|---|---|
| 1 | 5 | 0.3M | 37.32 |
| 2 | 10 | 0.8M | 30.71 |
| 3 | 20 | 2.1M | 27.69 |
| 4 | 40 | 4.8M | 27.38 |
| 5 | 80 | 10.2M | 26.46 |
| 6 | 160 | 21.1M | 27.38 |

Table 3: **Ultra-deep fractal networks** (CIFAR-100++). Increasing depth greatly improves accuracy until eventual diminishing returns. Contrast with plain networks, which are not trainable if made too deep (Table 4).

| Model | Depth | Train Loss | Error (%) |
|---|---|---|---|
| Plain | 5 | 0.786 | 36.62 |
| Plain | 10 | 0.159 | 32.47 |
| Plain | 20 | 0.037 | 31.31 |
| **Plain** | **40** | **0.580** | **38.84** |
| Fractal Col #1 | 5 | 0.677 | 37.23 |
| Fractal Col #2 | 10 | 0.141 | 32.85 |
| Fractal Col #3 | 20 | 0.029 | 31.31 |
| **Fractal Col #4** | **40** | **0.016** | **31.75** |
| **Fractal Full** | **40** | **0.015** | **27.40** |

Table 4: **Fractal structure as a training apparatus** (CIFAR-100++). Plain networks perform well if moderately deep, but exhibit worse convergence during training if instantiated with great depth. However, as a column trained within, and then extracted from, a fractal network with mixed drop-path, we recover a plain network that overcomes such depth limitation (possibly due to a student-teacher effect).

We run for 400 epochs on CIFAR, 20 epochs on SVHN, and 70 epochs on ImageNet. Our learning rate starts at 0.02 (for ImageNet, 0.001) and we train using stochastic gradient descent with batch size 100 (for ImageNet, 32) and momentum 0.9. For CIFAR/SVHN, we drop the learning rate by a factor of 10 whenever the number of remaining epochs halves. For ImageNet, we drop by a factor of 10 at epochs 50 and 65. We use Xavier initialization (Glorot & Bengio, 2010).

A widely employed (Lin et al., 2013; Clevert et al., 2016; Srivastava et al., 2015; He et al., 2016a;b; Huang et al., 2016b; Targ et al., 2016) scheme for data augmentation on CIFAR consists of only horizontal mirroring and translation (uniform offsets in $[-4, 4]$), with images zero-padded where needed after mean subtraction. We denote results achieved using no more than this degree of augmentation by appending a "+" to the dataset name (*e.g.* CIFAR-100+). A "++" marks results reliant on more data augmentation; here exact schemes may vary. Our entry in this category is modest and simply changes the zero-padding to reflect-padding.

## 4.2 RESULTS

Table 1 compares performance of FractalNet on CIFAR and SVHN with competing methods. FractalNet (depth 20) outperforms the original ResNet across the board. With data augmentation, our CIFAR-100 accuracy is close to that of the best ResNet variants. With neither augmentation nor regularization, FractalNet's performance on CIFAR is superior to both ResNet and ResNet with stochastic depth, suggesting that FractalNet may be less prone to overfitting. Most methods perform similarly on SVHN. Increasing depth to 40, while borrowing some parameter reduction tricks (Iandola et al., 2016), reveals FractalNet's performance to be consistent across a range of configuration choices.

Experiments without data augmentation highlight the power of drop-path regularization. On CIFAR-100, drop-path reduces FractalNet's error rate from $35.34\%$ to $28.20\%$. Unregularized ResNet is far behind ($44.76\%$) and ResNet with stochastic depth ($37.80\%$) does not catch up to our unregularized starting point of $35.34\%$. CIFAR-10 mirrors this story. With data augmentation, drop-path provides a boost (CIFAR-10), or does not significantly influence FractalNet's performance (CIFAR-100).

Note that the performance of the deepest column of the fractal network is close to that of the full network (statistically equivalent on CIFAR-10). This suggests that the fractal structure may be more important as a learning framework than as a final model architecture.

Table 2 shows that FractalNet scales to ImageNet, matching ResNet (He et al., 2016a) at equal depth. Note that, concurrent with our work, refinements to the residual network paradigm further improve the state-of-the-art on ImageNet. Wide residual networks (Zagoruyko & Komodakis, 2016) of 34-layers reduce single-crop Top-1 and Top-5 validation error by approximately $2\%$ and $1\%$, respectively, over

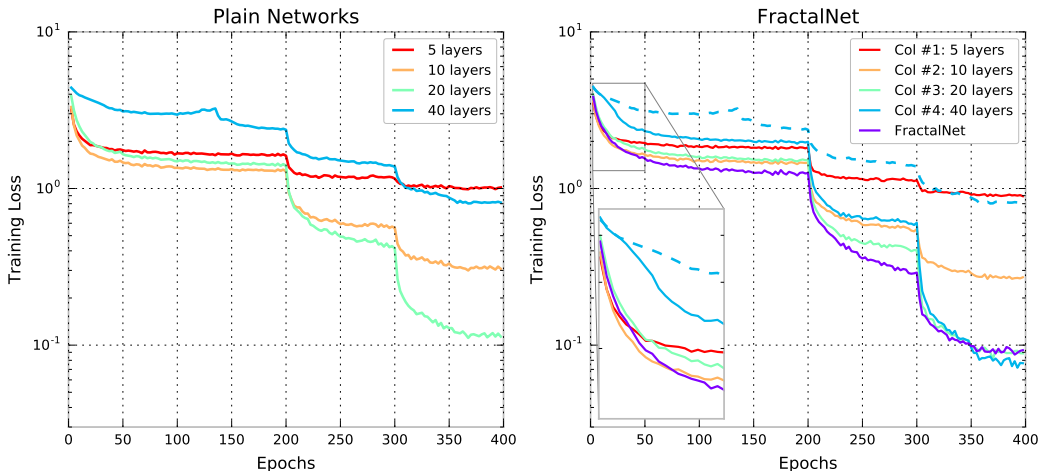

Figure 3: **Implicit deep supervision.** *Left:* Evolution of loss for plain networks of depth 5, 10, 20 and 40 trained on CIFAR-100. Training becomes increasingly difficult for deeper networks. At 40 layers, we are unable to train the network satisfactorily. *Right:* We train a 4 column fractal network with mixed drop-path, monitoring its loss as well as the losses of its four subnetworks corresponding to individual columns of the same depth as the plain networks. As the 20-layer subnetwork starts to stabilize, drop-path puts pressure on the 40-layer column to adapt, with the rest of the network as its teacher. This explains the elbow-shaped learning curve for Col #4 that occurs around 25 epochs.

ResNet-34 by doubling feature channels in each layer. DenseNets (Huang et al., 2016a) substantially improve performance by building residual blocks that concatenate rather than add feature channels.

Table 3 demonstrates that FractalNet resists performance degradation as we increase $C$ to obtain extremely deep networks (160 layers for $C = 6$). Scores in this table are not comparable to those in Table 1. For time and memory efficiency, we reduced block-wise feature channels to $(16, 32, 64, 128, 128)$ and the batch size to 50 for the supporting experiments in Tables 3 and 4.

Table 4 provides a baseline showing that training of plain deep networks begins to degrade by the time their depth reaches 40 layers. In our experience, a plain 160-layer completely fails to converge. This table also highlights the ability to use FractalNet and drop-path as an engine for extracting trained networks (columns) with the same topology as plain networks, but much higher test performance.

## 4.3   INTROSPECTION

With Figure 3, we examine the evolution of a 40-layer FractalNet during training. Tracking columns individually (recording their losses when run as stand-alone networks), we observe that the 40-layer column initially improves slowly, but picks up once the loss of the rest of the network begins to stabilize. Contrast with a plain 40-layer network trained alone (dashed blue line), which never makes fast progress. The column has the same initial plateau, but subsequently improves after 25 epochs, producing a loss curve uncharacteristic of plain networks.

We hypothesize that the fractal structure triggers effects akin to deep supervision and lateral student-teacher information flow. Column #4 joins with column #3 every other layer, and in every fourth layer this join involves no other columns. Once the fractal network partially relies on the signal going through column #3, drop-path puts pressure on column #4 to produce a replacement signal when column #3 is dropped. This task has constrained scope. A particular drop only requires two consecutive layers in column #4 to substitute for one in column #3 (a mini student-teacher problem).

This explanation of FractalNet dynamics parallels what, in concurrent work, Greff et al. (2017) claim for ResNet. Specifically, Greff et al. (2017) suggest residual networks learn unrolled iterative estimation, with each layer performing a gradual refinement on its input representation. The deepest FractalNet column could behave in the same manner, with the remainder of the network acting as a scaffold for building smaller refinement steps by doubling layers from one column to the next.

These interpretations appear not to mesh with the conclusions of Veit et al. (2016), who claim that ensemble-like behavior underlies the success of ResNet. This is certainly untrue of some very deep networks, as FractalNet provides a counterexample: we can extract a single column (plain network topology) and it alone (no ensembling) performs nearly as well as the entire network. Moreover, the gradual refinement view may offer an alternative explanation for the experiments of Veit et al. (2016). If each layer makes only a small modification, removing one may look, to the subsequent portion of the network, like injecting a small amount of input noise. Perhaps noise tolerance explains the gradual performance degradation that Veit et al. (2016) observe when removing ResNet layers.

## 5 CONCLUSION

Our experiments with fractal networks provide strong evidence that path length is fundamental for training ultra-deep neural networks; residuals are incidental. Key is the shared characteristic of FractalNet and ResNet: large nominal network depth, but effectively shorter paths for gradient propagation during training. Fractal architectures are arguably the simplest means of satisfying this requirement, and match residual networks in experimental performance. Fractal networks are resistant to being too deep; extra depth may slow training, but does not impair accuracy.

With drop-path, regularization of extremely deep fractal networks is intuitive and effective. Drop-path doubles as a method of enforcing speed (latency) vs. accuracy tradeoffs. For applications where fast responses have utility, we can obtain fractal networks whose partial evaluation yields good answers.

Our analysis connects the internal behavior of fractal networks with phenomena engineered into other networks. Their substructure resembles hand-crafted modules used as components in prior work. Their training evolution may emulate deep supervision and student-teacher learning.

### ACKNOWLEDGMENTS

We gratefully acknowledge the support of NVIDIA Corporation with the donation of GPUs used for this research.

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
