# Peer review of "FractalNet: Ultra-Deep Neural Networks without Residuals"

_ICLR 2017 — accepted_

[Reviewer Comment · AnonReviewer1 · 03 Dec 2016]
**parameters and comparison**

- Number of parameters needed is dramatically increasing with respect to competitors for state of art results. Can you also give comparison on depth, parameters,training times to Resnet and its variants in Table 1 and Table 2 and elaborate on this?
 Did you also try to decrease the number of parameters of 20 layers ?

-In Table 1, why didn't you report any result for no augmentation for 40 layers?

- Why did you set B values and corresponding layers and C values to given numbers, what else have you tried and what did you get in terms of training efficiency and misclassification error for getting the results on Table 1 and Table 2? Are the results stable or very sensitive according to these parameters? 

- Can you elaborate on the cases where you could not getter better results than best res-net variant in Table 1?

[Official Review · AnonReviewer2 · rating 6 · confidence 5 · 16 Dec 2016 (modified: 20 Jan 2017)]
**Unconvincing experimental comparisons**

This paper proposes a design principle for computation blocks in convolutional networks based on repeated application of expand and join operations resulting in a fractal-like structure. 

This paper is primarily about experimental evaluation, since the objective is to show that a residual formulation is not necessary to obtain good performance, at least on some tasks.

However, in my opinion the evaluations in the paper are not convincing. The primary issue is lack of a proper baseline, against which the improvements can be clearly demonstrated by making isolated changes. I understand that for this paper such a baseline is hard to construct, since it is about a novel architecture principle. This is why more effort should be put into this, so that core insights from this paper can be useful even after better performing architectures are discovered.
The number of parameters and amount of computation should be used to indicate how fair the comparisons are between architectures. Some detailed comments:

- In Table 1 comparisons to Resnets, the resnets from He et al. 2016b and Wide Resnets should be compared to FractalNet (in lieu of a proper baseline). The first outperforms FractalNet on CIFAR-100 while the second outperforms it on both. The authors compare to other results without augmentation, but did not perform additional experiments without augmentation for these architectures.

- The 40 layer Fractal Net should not be compared to other models unless the parameter reduction tricks are utilized for the other models as well.

- A proper comparison to Inception networks should also be performed for these networks. My guess is that the reason behind a seemingly 'ad-hoc' design of Inception modules is to reduce the computational footprint of the model (which is not a central motivation of fractal nets). Since this model is directly related to the Inception module due to use of shorter and longer paths without shortcuts, one can easily simplify the Inception design to build a strong baseline e.g. by converting the concatenation operation to a mean operation among equally sized convolution outputs. As an aside, note that Inception networks have already shown that residual networks are not necessary to obtain the best performance [1].

- It should be noted that Residual/Highway architectures do have a type of anytime property, as shown by lesioning experiments in Srivastava et al and Viet et al.

- The architecture specific drop-path regularization is interesting, but is used along with other regularizers such as dropout, batch norm and weight decay and its benefit on its own is not clear.

Overall, it's not clear to me that the experiments clearly demonstrate the utility of the proposed architecture. 

[1] Szegedy, Christian, Sergey Ioffe, and Vincent Vanhoucke. "Inception-v4, inception-resnet and the impact of residual connections on learning." arXiv preprint arXiv:1602.07261 (2016).

[Official Review · AnonReviewer1 · rating 5 · confidence 5 · 19 Dec 2016]
**Unsatisfactory experiments and restrictively large number of parameters**

This paper proposes a new architecture that does not explicitly use residuals but constructs an architecture that is composed of networks with fractal structure by using expand and join operations. Using the fractal architecture,  authors argue and try to demonstrate that the large nominal network depth with many short paths is the key for 'training 'ultra-deep” networks while residuals are incidental.

The main bottleneck of this paper is that number of parameters needed for the FractalNet is significantly higher than the baselines which makes it hard to scale to ''ultra-deep” networks.  Authors replied that Wide ResNets also require many parameters but this is not the case for ResNet and other ResNet variants. ResNet and ResNet with Stochastic depth scales to depth of 110 with 1.7M parameters and to depth of 1202 with 10.2M parameters which is much less than the number of parameters for depths of 20 and 40 in Table 1(Huang et al, 2016a).   It is not clear whether FractalNet can perform better than these depths with a reasonable computation. Authors report less parameters for 40 layers but this scaling trick is not validated for other depths including depth 20 in Table 1. On the other hand, the number of parameters for 40 layers with scaling trick is clearly still large compared to most of the baselines. Unsatisfactory comparison to these baselines makes the claims of authors unconvincing.

Authors also claim that drop-path to provide improvement compared to layer dropping procedure in Huang et al, 2016b however the results show that the empirical gain of this specific regularization disappears when well-known data augmentation techniques applied. Therefore the empirical effectiveness of drop-path is not convincing too.

DenseNets (Huang et al, 2016a) should be also included in the comparison since it outperforms most of the state of art Res Nets on both CIFAR10 and ImageNet and more importantly outperforms the proposed FractalNet significantly and it requires significantly less computation. 

Table 1 has Res-Net variants as baselines however Table 2 has only ResNet.  Therefore ImageNet comparison only shows that one can run FractalNet on ImageNet and can perform comparably well to ResNet which is not a satisfactory result given the improvements of other baselines over ResNet.  In addition, there is no improvement in SVHN dataset results and this is not discussed in the empirical analysis.

Also, authors give a list of some improvements over Inception (Szegedy et al., 2015) but again these intuitive claims about effectiveness of these changes are not supported with any empirical analysis. 

Although the paper attempts to explore many interesting intuitive directions using the proposed architecture, the empirical results are not support the given claims and the large number of parameters makes the model restrictive in practice hence the contribution does not seem to be significant. 

Pros:
Provides an interesting architecture compared to ResNet and its variants and investigates the differences to residual networks which can stimulate some other promising analysis

cons:
     -    Number of parameters are very large compared to baselines that can have even much higher depths with smaller number of parameters
The claims are intuitive but not supported well with empirical evidence
Path regularization does not yield improvement when the data augmentation is used
     -     The empirical results do not show whether the method is promising for “ultra-deep” networks

[Official Review · AnonReviewer3 · rating 6 · confidence 4 · 22 Dec 2016]
**weak comparison**

This paper presents a strategy for building deep neural networks via rules for expansion and merging of sub-networks.
pros:
- the idea is novel
- the approach is described clearly
cons:
- the experimental evaluation is not convincing, e.g. no improvement on SVHN
- number of parameters should be mentioned for all models for fair comparison
- the effect of drop-path seems to vanish with data augmentation

[Author Response · Michael Maire · 13 Jan 2017]
**Rebuttal**

We thank the reviewers for their time, but disagree with many points in the reviews.  We first address what we feel is the overall disconnect between the reviews and the contributions of the paper, then rebut specific points below.

Machine learning research is a mix of engineering, mathematics, and science.  Complete focus on engineering can restrict one to the narrow view of judging work by counting parameters and fractions of a percent in accuracy.  The reviews unfortunately reflect this mode of thinking.  However, long-term progress also requires scientific advancement, including new ideas as well as experiments that enhance understanding by revealing simple principles underlying complex phenomena.

FractalNet is a new idea with:

(1) Explanatory power:

As stated in the abstract, our experiments show that for the success of very deep networks, the "key may be the ability to transition, during training, from effectively shallow to deep".  This is the property shared by ResNet and FractalNet.

Furthermore, this observation is an important counterpoint to [Veit et al., NIPS 2016] which claims that: (a) ensemble-like behavior is key and (b) the fraction of available effectively short paths is somehow related to performance.  FractalNet provides a counterexample: we can extract a single column (plain network topology with one long path) and it alone (no ensembling) performs as well the entire network.  In FractalNet, the rest of the network is a training apparatus for transitioning from shallow to deep.  This strongly suggests that, with more careful analysis, the same may be true for ResNet.

In fact, unlike [Veit et al., NIPS 2016] our explanation is consistent with the view that very deep networks, such as ResNet and FractalNet, learn unrolled iterative estimation [Greff et al., Highway and Residual Networks learn Unrolled Iterative Estimation, arXiv:1612.07771 and ICLR 2017 submission].  In this view, the longest path is most important; the ensemble can be discarded after training.  Moreover, our Section 4.3 and Figure 3 provide insight into the dynamics of this learning process.

We believe that the research community values understanding these mechanisms and would benefit from the evidence our work injects into the debate.

(2) Simplifying power:

FractalNet shows how a simple design principle produces structure similar to the mishmash of hacks (hand-designed modules, manually-attached auxiliary losses at intermediate depths) required in prior architectures like Inception.  It also expands the set of simple tricks for building very deep networks to include an option other than residual connections.

(3) Good performance:

FractalNet matches a ResNet of equal depth in terms of performance on ImageNet.  Training either of these networks from scratch takes weeks on modern GPUs.  Yet, because FractalNet uses slightly more parameters or does not beat the absolute best ResNet variant (developed after 7 or so papers iterating on ResNet) on the smaller and less important CIFAR dataset, the reviews declare experiments unsatisfactory.  There is a line between demanding high experimental standards and strangling promising ideas; we are all for the former, but hope not to fall victim to the latter.

(4) New capabilities:

Contrary to AnonReviewer2's claim, prior work does not demonstrate an anytime property for ResNet; see our specific response below.  FractalNet, in combination with drop-path training, provides a novel anytime output capability, which could prove useful in real-world latency sensitive applications.


-----
Response to AnonReviewer1

Please see our comments about experiments above.

> Therefore the empirical effectiveness of drop-path is not convincing too.

Drop-path serves two purposes: (a) additional regularization in the absence of data augmentation and (b) regularization that allows the resulting network to have the anytime property.  Yes, additional data augmentation can compensate for drop-path if one only cares about (a), but drop-path is essential for and the only effective means of enabling anytime output.

> DenseNets (Huang et al, 2016a) should be also included in the comparison

DenseNet cites FractalNet as the original version of FractalNet (arXiv:1605.07648) was published on arXiv three months prior to the original version of DenseNet (arXiv:1608.06993).  We are happy to alert readers to subsequent work, but please keep in mind the historical sequence of development.

> Table 1 has Res-Net variants as baselines however Table 2 has only ResNet.

Many of the variants only reported results on CIFAR/SVHN.  For example, the Wide Residual Networks paper (arXiv:1605.07146), at the time of the ICLR deadline (November 5), did not include ImageNet results.  It was updated weeks later, on November 28, to include ImageNet results.  We can expand the table, but it is a bit difficult to have already compared to results that did not exist at submission time.

> there is no improvement in SVHN dataset results and this is not discussed in the empirical analysis.

Section 4.2 observes "most methods perform similarly on SVHN".  The errors rates on SVHN are so low as to be uninformative.  We believe SVHN is simply too easy of a dataset to be a challenge to any of the modern techniques in Table 1.  Much like MNIST, SVHN is now a useful sanity check, but not a differentiator.

> Also, authors give a list of some improvements over Inception (Szegedy et al., 2015) but again these intuitive claims about effectiveness of these changes are not supported with any empirical analysis.

Our point is merely to draw connections between the FractalNet architecture and some hand-designed components of Inception, suggesting a more fundamental explanation for why those particular hand-designed choices are effective.  Widespread adoption of ResNet by the community has displaced Inception, VGG, and other architectures, making ResNet the clear leader and appropriate target for experimental comparison.


-----
Response to AnonReviewer2

Please see our comments about experiments and the main contributions of the paper above.

> The 40 layer Fractal Net should not be compared to other models unless the parameter reduction tricks are utilized for the other models as well.

We are demonstrating feasibility of parameter reduction tricks for FractalNet.  Much of the iterative improvement in the ResNet variants themselves, developed over many papers, already has a component of optimizing the architecture to reduce parameters and/or computational load.

> Since this model is directly related to the Inception module due to use of shorter and longer paths without shortcuts.

This statement is incorrect.  Our model is not directly related to Inception.  It reproduces some local connectivity reminiscent of Inception modules, but the global connectivity structure is entirely different. Even locally, FractalNet has a recursive join structure which Inception modules lack.  As consequence of this is that at global scale, FractalNet contains many paths whose length range over many orders of magnitude (all powers of 2 up to the maximum depth).  In contrast, the shortest path through an Inception network grows linearly with depth (number of modules stacked).

> As an aside, note that Inception networks have already shown that residual networks are not necessary to obtain the best performance

ImageNet performance is a useful benchmark tool, not the end goal of network design.  Mass adoption of ResNet by the community suggests that simplicity and scalability are also legitimate concerns.  With regard to scalability in depth, the only networks previously demonstrated to easily extend to the 100-1000 layer regime rely on some form of residual connection (ResNet/Highway Networks).  We demonstrate a 160-layer FractalNet, placing it in this group.

Moreover, FractalNet and ResNet/Highway all have a mechanism for evolving from being effectively shallow to effectively deep over the course of training. This shallow to deep evolution seems critical, but unfortunately the design of Inception prevents its effective depth from being less than the number of stacked Inception modules.

> It should be noted that Residual/Highway architectures do have a type of anytime property, as shown by lesioning experiments in Srivastava et al and Viet et al.

We disagree.  The lesioning experiments in Veit et al. demonstrate the opposite: ResNet does not have an anytime property.  If deleting a few layers, then yes, ResNet can recover.  However, Section 4.2 and Figure 5 of Veit et al, show that deleting 10 blocks of a 54 block ResNet increases CIFAR-10 error to 0.2, deleting 20 blocks pushes error above 0.5.  So one cannot maintain a reasonable error if halving the ResNet depth.  Compare this to our Table 4: subnetwork columns of one half (20) and even one fourth (10) the layers of a full FractalNet (40) maintain low error on CIFAR-100.  This robustness across many orders of magnitude in depth (and thus time) is required for a useful anytime property and is unique to FractalNet (with drop-path training).

> The architecture specific drop-path regularization is interesting, but is used along with other regularizers such as dropout, batch norm and weight decay and its benefit on its own is not clear.

Drop-path is essential for and the only effective means of enabling anytime output.  The fact that it can contribute as a regularizer in other scenarios is a bonus.


-----
Response to AnonReviewer3

> the experimental evaluation is not convincing, e.g. no improvement on SVHN

As we replied to AnonReviewer1: Section 4.2 observes "most methods perform similarly on SVHN".  The errors rates on SVHN are so low as to be uninformative.  We believe SVHN is simply too easy of a dataset to be a challenge to any of the modern techniques in Table 1.  Much like MNIST, SVHN is now a useful sanity check, but not a differentiator.

> number of parameters should be mentioned for all models for fair comparison

We can add this to the table, but please see our larger discussion of experiments above.

> the effect of drop-path seems to vanish with data augmentation

Drop-path is essential for and the only effective means of enabling anytime output.  The fact that it can contribute as a regularizer in the absence of data augmentation is a bonus.

[Public Comment · George Philipp · rating 7 · confidence 3 · 17 Jan 2017]
**From an interested reader: I agree with the authors rebuttals**

Looking through the comment section here, I agree to a large degree with the author's standpoint on many issues discussed. Points (1) through (4) in the authors comment below are, in my opinion, a good summary of the contributions of the paper. While I don't think those contributions are groundbreaking, I believe they are significant enough to merit acceptance.

The reason I am commenting here is because, having looked at several comment sections for this ICLR, I am seeing a general trend that reviews have a strong focus on performance, i.e. reviews tend to be very short and judge papers, to a large degree, on whether they are a few percentage points better or worse than the reported baseline. E.g. see the comments "the experimental evaluation is not convincing, e.g. no improvement on SVHN" or "the effect of drop-path seems to vanish with data augmentation" below.

I believe that papers should be judged more on their scientific contributions (see points (1), (2) and (4) below), especially when those papers themselves state that their focus is on those scientific contributions, not on amazing performance.

Further, I believe the trend to focus excessively on performance is problematic for a number of reasons:

 - The Deep Learning community has focused very heavily on a few datasets (MNIST, ImageNet, CIFAR-10, CIFAR-100, SVHN). This means that at any time, a large chunk of the deep learning literature is battling for 5 SOTA titles. Hence, expecting any new model to attain one of those titles is a very high bar.

 - It is an arbitrary standard. Say the SOTA on ImageNet improves by 2% a year. Then a paper that outperforms by 1% in 2014 would underperform by 1% in 2015. By the performance standard, the same paper with the same ideas and the same scientific merit would have declined drastically in value over that one year. Is that really true?

 - How does one even draw a "fair comparison" on these standard datasets at this point? The bag of tricks for neural networks includes: drop-out, l2, l1, ensembling, various forms of data augmentation, various forms of normalization and initialization, various non-linearities, various learning rate schedules, various forms of pooling, label smoothing, gradient clipping etc. etc. There are a gazillion ways to eke out fractions of percentage points of performance. And - every single paper has a unique combination of tricks that they use for their model, even though the tricks themselves are unrelated to the model. Hence, the only truly fair comparison would be to compare against every reference model with the exact trick combination that the paper presenting the reference model used, which would take an exorbitant amount of time. What's worse, many papers do not even report all of the tricks they used. One would have to get the authors code and reverse engineer the model, not to mention slight differences introduced by using e.g. TensorFlow vs. Torch vs. Caffe. In this light, the request from one of the reviewers to have a baseline "against which the improvements can be clearly demonstrated by making isolated changes" seems unrealistic to me.

 - The ML community should not make excessive fine-tuning of models mandatory for publication. By requiring models to beat SOTA, we force each author to fine-tune their model ad nauseum, which leads to an arms race. To get a publications, authors would spend ever more time fine-tuning their models. This can not only lead to "training on the test set", but also wastes the time of researcher that could be better spent exploring new ideas.

 - It gives too much power to bad research. In science, there is always a certain background rate of "bad" results published: either the numbers are outright fake or the experimental protocol was invalid, e.g. someone used the test set as a validation set or someone did an exorbitant number of random reruns and only published the best single result. What's worse, these "bad" results are far more likely to hold the SOTA title at any given time than a "good" result. By requiring new publications to beat SOTA, we give too much power to bad results.

 - It punishes authors for reporting many or strong baselines. In this paper, authors were careful to report many recent results. Table 1 is thorough. And now they are criticized for not beating all of those baselines. I have a feeling that if the authors of this paper had been more selective about which baselines they report, i.e. those that they can beat, they would have received higher scores on the paper. I have written an in-depth review for another paper at this conference that used, in my opinion, very weak baselines and ended up getting high reviewer marks. I don't think that was a coincidence. 

The same arguments apply, though I think to a lesser degree, to judging models excessively on how many parameters they have or their runtime. However, I agree with reviewers that more information about how models compare in terms of those metrics would enhance this paper. I would like to see a discussion of that in the final version. In general, I think this paper would benefit from an appendix with more details on model and training procedure. I also agree with reviewers that 80 layers, which is the deepest that authors can go while improving test error (Table 3), is not ultra-deep. Hence putting "ultra-deep" in the paper title seems exaggerated and I would recommend scaling back the language. However, I don't think being ultra-deep (~1000 layers) is necessary, because as Veit et al showed, networks that appear ultra deep might not be ultra deep in practice. Training an 80-layer net that functions at test time without residual connections seems to be enough of an achievement.

In summary, I think if a paper makes scientific contribution (see points (1), (2) and (4) below) independent of performance, then competitive performance should be enough for publication, instead of requiring SOTA. I believe this paper achieves that mark.

[Reviewer Comment · AnonReviewer1 · 23 Jan 2017]
**rebuttal response**

Authors claim that the reviewers reflect "narrow view" of on the number of parameters and fractions of accuracy and  "complete focus on engineering" etc . Authors claims about focusing on engineering are open to debate as only the technicality in the paper is the fractal network architecture with intuitive claims. One can either formulate the system with rigorous analysis with clear assumptions (which would require satisfactory theoretical analysis and relatively small scale empirical sanity check without heavy experiments) or propose a comprehensive empirical analysis by designing careful experiments to support intuitive claims. Therefore, as the approach clearly does not belong to the first category,  such an approach needs strong experimental evidence to support intuitive claims without a doubt.  The rebuttal lists unsatisfactory answers with somehow manipulative arguments about narrow reviewing/ dates of the baselines( which will be discussed below).. 


Authors state in the paper that " In experiments, fractal networks **match the state-of-the-art performance held by residual networks** on both CIFAR and ImageNet classification tasks, thereby demonstrating that residual representations may not be fundamental to the success of extremely deep convolutional neural networks" and "Fractal architectures are arguably the simplest means of satisfying this requirement, and match or exceed residual networks in experimental performance". Therefore the main support behind all the claims is that fractal network match (or exceed) performance of state of art residual networks, indeed the empirical study only compares accuracy to some baselines without designing empirical analysis for the claims about differences to resnets (or ensemble explanation of resnet by Veit et al(NIPS 2016) ). 
However even this comparison lacks many results as explained in my review.   Therefore the expectation of the fair comparison is completely reasonable and it is not about only focusing on fractions of accuracy or number of parameters. As a good example, Veit et al demonstrates systematical empirical support for their claims about analyzing residual networks (which seems to be on arxiv since May but still not cited on paper ). Table 4 and Figure 3 provide good preliminary sanity check but compares only to plain networks, do not support claims about differences between residual and fractal networks.

The paper claims that the fractal network scales to "ultra" deep networks, however authors can not report results on dozens of layers. Authors also claim that extra depth may slow training (not clear how much) but does not decrease accuracy but this is not clear as there are no results for dozens of layers, also in Table 3, error increases as depth increase to 160 layers.  Number of parameters of the proposed architecture is significantly greater than the state of art resnet variants as I explained in the review but the authors argue that it is slightly more parameters. 

As an answer to lack of comparison to DenseNet, authors argue in rebuttal that DenseNet cites FractalNet but this can not be a reason for the lack of comparison as DenseNet was published in August and well-known as holding state of art results as a resnet variant. 

Authors state in rebuttal that "Many of the variants only reported results on CIFAR/SVHN… it is a bit difficult to have already compared to results that did not exist at submission time." .  However there were clearly published ones, for example two results by Huang et al, 2016b on july (arxiv) ..

Regarding the "simplifying power" claim of authors.  It is not very convincing that it is simplifying. how can it simplify with a harder training procedure with many parameters that can not scale as good as baselines?

In essence, my evaluation did not change as rebuttal did not provide satisfactory clarification or improvement.

[Final Decision · Program Chairs · 06 Feb 2017]
**ICLR committee final decision**

The paper describes a novel and simple way of constructing deep neural networks using a fractal expansion rule. It is evaluated on several datasets (ImageNet, CIFAR 10/100, SVHN) with promising results which are on par with ResNets. As noted by the reviewers, FractalNets would benefit from additional exploration and analysis.